# Planar Optofluidic Integration of Ring Resonator and Microfluidic Channels

**DOI:** 10.3390/mi13071028

**Published:** 2022-06-28

**Authors:** Genni Testa, Gianluca Persichetti, Romeo Bernini

**Affiliations:** Institute for Electromagnetic Sensing of the Environment (IREA), National Research Council (CNR), Via Diocleziano, 328, 80124 Naples, Italy; persichetti.g@irea.cnr.it (G.P.); bernini.r@irea.cnr.it (R.B.)

**Keywords:** optical resonators, optofluidics, integrated optics, antiresonant reflecting optical waveguide (ARROW), microfluidics

## Abstract

We report an optofluidic hybrid silicon-polymer planar ring resonator with integrated microfluidic channels for efficient liquid delivery. The device features a planar architecture of intersecting liquid-core waveguides and microfluidic channels. A low-loss integration of microfluidic channels is accomplished by exploiting the interference pattern created by the self-imaging effect in the multimode interference-based coupler waveguides. Numerical simulations have been performed in order to minimize the propagation losses along the ring loop caused by the integration of microfluidic channels. The device has been fabricated and optically characterized by measuring the quality factor, obtaining a value of 4 × 10^3^. This result is comparable with the quality factor of an optofluidic ring with the same optical layout but without integrated microfluidic channels, thus, confirming the suitability of the proposed approach for microfluidics integration in planar optofluidic design.

## 1. Introduction

Optofluidics have emerged in recent years as a new discipline able to combine fluidic and optical functions on a miniaturized scale in a unique and unprecedented manner [1,2,3]. Thanks to the recent advances in the field of microfluidics, simpler and precise handling of fluids can be achieved on a microscale, offering new opportunities to realize optofluidic devices [4]. In addition to the advantage of reducing the overall device size, microfluidic integration can improve the repeatability of the device response by allowing for fine control of the liquid flow.

In the field of optical sensing, microfluidic integration presents the relevant implication of reducing sample and reagent volumes and of preventing evaporation of the sample under analysis during tests. Due to the enormous benefits provided by the optofluidic approach in the realization of miniaturized optical sensors, a huge number of sensor architectures have been proposed in recent years for the realization of self-containing sensors for liquid sensing, the so-called optical “lab-on-a-chip” (LOC) [1,5,6]. 

The prospect of lowering the cost and simplifying the fabrication procedure has made polymer materials the favorite choice for building microfluidic systems, in turn, inspiring the development of optofluidic devices based on hybrid integration of polymer microfluidic and silicon optical modules assembled with a multilayered modular approach [7,8,9,10]. 

Waveguides integrated with microfluidic channels often form the basis of microfluidic networks for optofluidic device design [11,12,13,14]. Integrated devices which combine both fluidic channels and optical waveguides on a planar platform have been demonstrated [15,16,17]. 

The introduction of integrated optofluidic liquid-core waveguides represents a step forward in the development of optofluidic devices as they intrinsically provide devices with microfluidic capability [1,18]. However, microfluidic components such as the inlet and outlet and channels for injecting and guiding fluids need to be interconnected with optofluidic channels, without affecting the optical performance of the waveguides themselves.

Optofluidic waveguides can serve as sensors. Because the waveguide itself acts as a channel for transporting both the liquid sample and the optical probing mode, a high level of compactness and increased sensitivity can be achieved by implementing an optofluidic waveguide as a building block in sensor design. 

Silicon-based optofluidic waveguides such as slot, photonic crystal waveguides and ARROWs have shown great potential for planar photonics and novel optofluidic devices for sensing applications [10,18,19,20]. 

In the study by Measor et al. [21], an architecture of solid-core and optofluidic waveguides, comprising fluidic reservoirs for sample storage connected to the optofluidic waveguide, has been integrated on a planar optofluidic platform for sensing applications. In the study by Testa et al. [9], a planar optofluidic platform for sensing applications has been realized with a modular hybrid approach, with a microfluidic inlet, outlet and a micromixer fabricated in the top polymeric layers and fluidically connected with the underneath optofluidic waveguide. 

The integration of microfluidic connections has become more difficult in the case of micron-sized and high performance interferometric photonic devices, such as optofluidic ring resonators, where an unsuitable device design could cause a strong degradation of the device performance. 

In the study by Chandrahalim et al. [22], fluidic channels for sample delivery in a planar silica-based hollow core ring resonator were inserted on the inner side of the hollow waveguides shaping the cavity where virtually no WGM is excited, without significantly affecting the optical performance of the resonator.

In previous work, we presented a hybrid planar optofluidic ring resonator sensor [23], where each of the photonic functional elements (the cavity, the bus waveguide and the coupler) was formed by liquid-core waveguides (hybrid ARROW) [24]. The proposed ring did not integrate any microfluidic channels for sample delivery; thus, the liquid core was filled by capillary action from the open-ended bus waveguide, and the sensing performance was evaluated by thermally induced changes in the core refractive index. 

In a planar optofluidic ring resonator, microfluidic channels must be properly placed along the ring path in order to ensure uniform and rapid liquid replacement. However, inserting microfluidic channels across the optofluidic waveguides can increase the propagation losses by disrupting the optical confinement in the waveguides and decreasing the Q-factor.

In this study, we design, simulate and experimentally demonstrate that noninvasive substrate integration of a planar network of microfluidic channels in an ARRROW-based hybrid optofluidic ring resonator can be accomplished by carefully designing the device geometry. In particular, the integration of microfluidic channels is achieved by exploiting the interference pattern created by the self-imaging effect in multimode interference waveguide couplers (MMI). 

## 2. Design and Simulations

Figure 1a shows the basic ring layout. Optical design optimization of the ring has been performed by following the general rules as reported in [25]. Hybrid ARROW waveguides with four antiresonant cladding layers of alternating high and low refractive indexes shape the ring resonator (Figure 1b) [24]. The waveguide, core width d_c_ = 3 μm and depth h = 12 μm, is designed to support linearly polarized, quasi-single mode operation at the working wavelength of 780 nm. 

The ring resonator comprises the bus waveguides, the MMI coupler (MMI_c_), bend-shaped waveguides with a curvature radius of R = 100 μm and a single mode straight waveguide (SW). The MMI_c_ has a width of W_MMI_ = 10.24 μm and a length of L_MMI_ = 354 μm, and it is designed to equally split the input power into the two output waveguides (balanced splitter). The SW has a length of L = L_MMI_. 

For microfluidic channels (MCs) integration, two important points have been considered: (1) from a fluidic point of view, inlet and outlet channels should be opportunely positioned in order to ensure rapid and complete liquid replacement in the cavity, and (2) from an optical point of view, the channels should be positioned in order to minimize the induced disturbance on the propagating mode.

Regarding the last requirement, fluidic connections should be positioned on the input and output bus waveguides in order to avoid any perturbation of the ring cavity (see Figure 2a). However, from a fluidic point of view, this configuration can be modeled as two hydraulic resistances in parallel, R_MMI_ and R_Ring_, that are strongly unbalanced (R_Ring_ ≈ 60 R_MMI_) [26]. This implies a very slow liquid flow rate in the ring that, in turn, can cause inefficient replacement of the liquid. By placing the inlet and outlet channels on opposite sides of the ring loop (Figure 2b) it is possible to balance the fluidic resistances, R_L_ and R_R_, and ensure a fast and complete replacement of the liquid in the cavity. In this configuration we have estimated R_R_ ≈ 0.96 R_L_. 

In order to compare the fluidic schemes, finite element method (FEM) microfluidics simulations using COMSOL Multiphysics software were performed. Three-dimensional steady-state velocity distribution of the solution is shown in Figure 3 for both the unbalanced and balanced fluidic ring scheme. In both schemes, the flow rate of the inlet fluid stream was set to 0.1 m/s. 

Figure 3a reports the simulation result for the unbalanced configuration. The ratio R_Ring_/R_MMI_ can be calculated by taking the flow rate ratio FRR_U_ = Q_MMIc_/Q_SW_ between the flow rate in the MMIc (Q_MMIc_) and in the SW (Q_SW_) [26], obtaining the FRR_U_ = 77.3. In Figure 3b) the simulation result for the balanced configuration is reported. In this case, the ratio R_R_/R_L_ can be calculated as the flow rate ratio FRR_B_ = Q_L_/Q_R_ = 1.3. These results confirm that in the unbalanced fluidic scheme the liquid flows substantially in the MMIc, causing an inefficient liquid replacement in the ring loop. In the balanced fluidic scheme, instead, the inlet fluid flows along the whole ring cavity. Moreover, because the flow rate in the input and output bus waveguides is one order of magnitude lower than the rate of the flow in the ring cavity, the balanced ring configuration prevents the problem of liquid leakage from the open-ended bus waveguides.

On the basis of the fluidic analysis, the layout of the optofluidic ring resonator has been designed with microfluidic channels placed on the opposite side of the ring loop, following the scheme of Figure 2b. However, this configuration requires an accurate optical design to avoid an increase in the optical losses at the intersection point of the microfluidic channels with the waveguide wall of the ring cavity. 

Figure 4 shows the proposed layout. As can be observed from Figure 4, an additional multimode interference device (MMI_SW_) has been integrated to assist microfluidic channel insertion on the SW, as will be discussed in the following section. 

The effect on the propagation losses due to the removal of the waveguide boundary in correspondence with the MCs insertions has been investigated by numerical simulations. Numerical simulations have been performed by using a two-dimensional finite-difference time-domain method (2D FDTD) (OMNISIM, © Photon Design, Oxford, UK). The optical behavior of the MMI coupler (MMI_c_) and of the straight optofluidic waveguide channel has been studied.

First, the unperturbed MMI_c_ (without integrated MCs) has been simulated. The field of a single mode ARROW waveguide at λ = 780 nm is considered the exciting optical source on the input waveguide (i1). Figure 5 displays the interference pattern created by the multimode interference effect in the MMI_c_. From simulations, we have estimated the normalized power transmitted on output channel 1 (O_1_), $P_{1}=P_{O1}/P_{0}$, and output channel 2 (O_2_), $P_{2}=P_{O2}/P_{0}$, of the unperturbed MMI. $\;P_{O1\;}$ and $P_{O2}\;$ are the power transmitted from port O_1_ and port O_2_ and $P_{0}$ is the reference input power, transmitted from the input port i1 and acting as the exciting field. The transmitted power is referred to the fundamental ARROW mode. We obtained $P_{1}=0.4869$ and $P_{2}=0.4826$. The discrepancy with the theoretical values of *P*_1t_ = 0.50 and *P*_2t_ = 0.50 has already been described elsewhere [25] and can be explained by taking into account the optical losses due to the leaky propagation of the h-ARROW modes. Because the MMI effect is the result of the co-propagation of leaky higher order modes, interferential output images undergo degradation to some extent. From the Figure 5, the interference produces an intensity profile along the MMI_c_ length with a series of very low (virtually null) intensity regions in the proximity of the waveguide walls where microfluidic channels can be inserted to minimize their influence on the light propagation. It should be noted that during the light circulation in the ring cavity the MMI_c_ is alternately excited from both the input ports (O_1_, O_2_) and, in particular, by the bus waveguide (port O_1_) and the by the ring loop (port O_2_), in the former case producing an interference pattern that is mirrored with respect to the propagation axis z as compared with the pattern shown in the Figure 5. Based on this consideration, in order to reduce the perturbation on waveguide propagation, the microfluidic channel can be placed where a null intensity region appears in the proximity of both opposite MMI_c_ walls (shaded box in the figure).

Hence, the MMI_c_ has been simulated by inserting a 14 μm wide microfluidic channel which is located at z = 305 µm, where the field intensity decreases substantially near the ARROW waveguide wall. Numerical results give, in this case, transmitted powers $P_{1}=0.4872\;$ and $P_{2}=0.4825$. These values are very close to the ones calculated for the unperturbed MMI_c_ and demonstrate that the integration of the MC does not introduce additional losses, and the modal propagation is substantially unaffected by the integration of the microfluidic channel. In order to confirm the effectiveness of our design approach, the MMI_c_ has been simulated by including a 14 μm wide MC placed at z = 205 µm, where the field is high near the waveguide wall. In this case, we obtained $P_{1}=0.4352\;$ and $P_{2}=0.4537$, resulting in a power drop of about 10.6% and 6.8% on channels O_1_ and O_2_, respectively. The results are summarized in Table 1.

Concerning the insertion of the microchannel MC_2_ along the ring loop, we evaluated the impact of this integration along the straight waveguide (SW), that is a single mode h-ARROW waveguide, on the mode propagation. We preliminarily evaluated the normalized power transmitted by the unperturbed waveguide (no MC_2_), obtaining a value of $P_{out,ch\;}/P_{0,ch}=0.969$. *P_0,ch_* is the reference power, referred to the fundamental input ARROW mode. This result was then compared with the normalized power transmitted by the same waveguide with an integrated 10 µm-wide MC, which is centered with respect to the propagation axis of the SW (see Figure 2), obtaining $P_{out,ch}^{MC2}/P_{0,ch}=0.848$. Hence, a power drop of about 13% should be expected due to microfluidic integration. 

Using the same argument, the insertion of the outlet microfluidic channel was accomplished by adding an MMI device (MMI_SW_), properly designed and incorporated along the straight waveguide section SW. The MMI_SW_ is constituted by a couple of 1 × 1 MMI placed in a cascade along the propagation axis. 

A 1 × 1 MMI device transfers light from the input to the output waveguide with minimal loss. Because an ARROW-based MMI is intrinsically a lossy optical element, we considered the minimum width and length to create a suitable interference pattern. Starting from the waveguide width d_c_ and considering the fabrication constraints, 1 × 1 MMI sections with dimensions of width = 5 µm and length L_1_ = 42.6 µm could be considered. The interference pattern created by a 1 × 1 MMI section of length 2L_1_ is shown in Figure 6. As it can be observed from the figure, the interference created by a single 1 × 1 MMI section of length L_1_ does not exhibit very low-intensity regions near the waveguide walls, which are wide enough to locate the microfluidic channel. Instead, by doubling the MMI sections length it is possible to exploit the reimaging effect created by the first section of length L_1_ in the center of the MMI_SW_ to produce a strong decrease in the intensity near the waveguide wall and where the MC_2_ can be placed (shaded box of Figure 6). For this reason, the length of the MMI_SW_ has been set to L_MMISW_ = 2L_1_. 

By numerical simulations we found that the introduction of the MMI_SW_ causes a power drop of about 1.6% along the SW. The subsequent integration of the microfluidic channel MC_2_ leads to a total power drop of 2.4% as compared with the simple straight waveguide channel. By comparing this value with the power loss caused by mere integration of MC_2_ on the straight waveguide, equal to 13%, it could be observed that the proposed introduction of the MMI_SW_ allows a great enhancement of the optical performance of the considered ring section, as proposed. The simulation results of the MC_2_ integration are summarized in Table 2. 

Finally, we simulated the flow profile of the ring layout shown in Figure 4, optimized according to the obtained numerical results. The MC_1_ with a width of 14 µm is located at z = 305 along the z axis of the MMIc, and the MC_2_ with a width of 10 µm is in the center of the MMI_SW_. Figure 7 displays the velocity distribution, numerically simulated.

Moreover, in this case we have estimated the flow rate ratio between the flows in the opposite curved sections, obtaining the FRR = Q_L_/Q_R_ = 1.3.

Hence, fluidic and optical simulation results suggest that by exploiting the presence of the MMI_c_ and introducing an additional MMI device (MMI_SW_), we could successfully integrate microfluidic channels while keeping the induced propagation losses as low as possible. 

## 3. Experimental Results

### 3.1. Fabrication

The optofluidic ring resonator comprises a silicon-based part fabricated by bulk micromachining, following the procedure described in [23], and a polymer part, fabricated by soft lithography.

Briefly, the silicon part has been manufactured by bulk micromachining. Deep reactive-ion etching was used to create trenches of 4.76 μm width and 12.88 μm depth into silicon, and a low-pressure chemical vapor deposition (LPCVD) process was employed to deposit four alternating antiresonant cladding layers of silicon nitride (Si_3_N_4_ (*n* = 2.04), 124 nm thick) and silicon dioxide (SiO_2_ (*n* = 1.44), 315 nm thick). 

In Figure 8, a scanning electron microscope (SEM) image of the silicon-based part is shown. Two hole-markers with diameters of 100 µm have been included in the design in order to simplify the integration of the top polymeric microfluidic layer, as will be discussed in the following section. 

The top polymeric microfluidic layer has a thickness of 1 mm and was fabricated by cast and curing polydimethylsiloxane (PDMS) from a master mold of poly(methyl methacrylate) (PMMA) material. The PMMA master has been microstructured by using a computer numerical control (CNC) micromilling machine. 

A series of through-holes of 300 µm in diameter have been drilled by the CNC machine in the PMMA substrate (Figure 9a). The holes were suitably located such that they match the relative position of the hole markers in the silicon part. After that, polymide tubing with an outer diameter of 300 µm was inserted in the holes of the board in the PMMA master (Figure 9b). The as-prepared PMMA master was used as a mold for the PDMS casting process (Figure 9c). The master mold was filled with PDMS prepolymer and de-gassed under vacuum (Figure 9d). A total curing time of 1 h and a temperature of 100 °C were used. When the PDMS was partially cured (after 30 min from the start of the curing process), the polymide tubing was gently pulled from PDMS in order to come out from the holes in the PMMA, still remaining integrated with the PDMS layer. Once the curing process ended, the PDMS integrating the tubing was gently removed from the master (Figure 9e). 

The exploded view drawing of the chip constituted by the bottom silicon part and as-prepared PDMS top layer is shown in Figure 10a. In order to form the final chip, the PDMS layer and the silicon part were permanently bonded by an oxygen plasma-assisted bonding procedure. Both parts were exposed to plasma oxygen for 30 s at a pressure level of 0.4 mbar and power of 40 W. Subsequently, the two parts were put in contact by using a micrometer translation and rotation stage and a CCD camera to control the alignment (Figure 10b). Figure 10c presents the photograph of the assembled chip.

### 3.2. Optical Characterization 

We have experimentally characterized the optofluidic ring resonator with an integrated MC_s_ (ORR_M_) and evaluated the optical performances by measuring the quality factor Q. 

Q-factor can be derived from the transmitted spectrum by measuring the full width at half-maximum of the optical resonances. Figure 11 shows the experimental setup for optical characterization of the ORR_M_. Measurements were performed by coupling light from a tunable laser (TL, 760÷781 nm) in and out of the bus waveguide through optical fibers (OF, single mode at 780 nm). The collected light was measured with a photodiode (Pd) and analyzed with an oscilloscope (Os). To inject and collect liquid samples, Tygon tubing (TT) with an inner diameter of 300 µm was attached to the polymide tubing and the syringe pump (SP). 

To assess the effectiveness of our approach, we first measured the Q-factor of an optofluidic ring resonator with the same optical and geometrical architecture (h-ARROW, curvature radius R and MMI_c_ design) and fabricated under the same conditions but without any integrated microfluidic channels or MMI_SW_ (ORR) [23]. The experimental setup for the characterization of the ORR was the same as in Figure 11, except that the device core was filled by capillary action with water from the open-ended bus waveguide. 

Figure 12a shows the measured spectrum and b) the spectrum of a single optical resonance. The resonance has been fit to a Lorentzian curve. We measured a full width at half-maximum of about Δλ ≈ 196 pm and a Q-factor of 3.96 × 10^3^.

Figure 13 shows the optical resonance of the ORR_M_. In this case, water has been infiltrated into the device via the integrated microfluidic channels and inlet/outlet by using a syringe pump. From Figure 13, we measured a full width at half-maximum of about Δλres ≈ 194 pm, resulting in a Q-factor of 4 × 10^3^ at around 777.85 nm. This result compares well with the Q-factor of the ORR and confirms that with a strategic design of the entire device, we can integrate microfluidic channels while keeping the optical performance of the optofluidic ring almost unaltered. 

## 4. Conclusions

In conclusion, a planar integrated optofluidic ring resonator with integrated microfluidic channels for liquid injection has been successfully designed, fabricated and characterized. Because the optofluidic ring is shaped by liquid-core waveguides, planar integration of microfluidic channels has been accomplished by the innovative design of the optofluidic network. The relative in-plane positioning of the inlet and outlet microfluidic channels along the ring loop has been established by taking into account both the microfluidic configuration and the propagation losses of the ring sections where the channels need to be inserted. Numerical simulations of the ring sections have been performed with the aim of evaluating the optical losses induced by microfluidic channel insertion. We found that the multimode interference pattern created in the MMI_c_ can be exploited to minimize the propagation losses caused by the waveguides’ wall disruptions. Using the same argument, we added a further MMI (MMI_SW_) along the straight waveguide section, where the outlet channel needs to be inserted. 

The optofluidic ring, combined with carefully designed integrated microfluidic channels, has shown performances comparable with the bare version of the ring. In particular, a quality factor of Q ≈ 4 × 10^3^ has been experimentally demonstrated.

## Figures and Tables

**Figure 1 micromachines-13-01028-f001:**
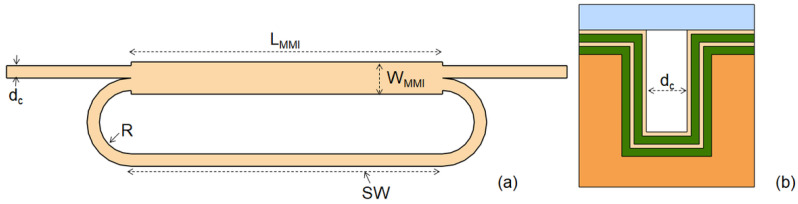
(**a**) Schematic layout of the planar optofluidic ring resonator based on h-ARROW. (**b**) Schematic cross section of an h-ARROW.

**Figure 2 micromachines-13-01028-f002:**
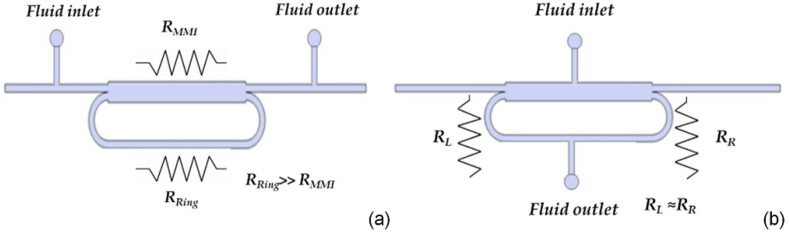
Possible schemes for microfluidic integration. (**a**) Unbalanced and (**b**) balanced fluidic resistances.

**Figure 3 micromachines-13-01028-f003:**
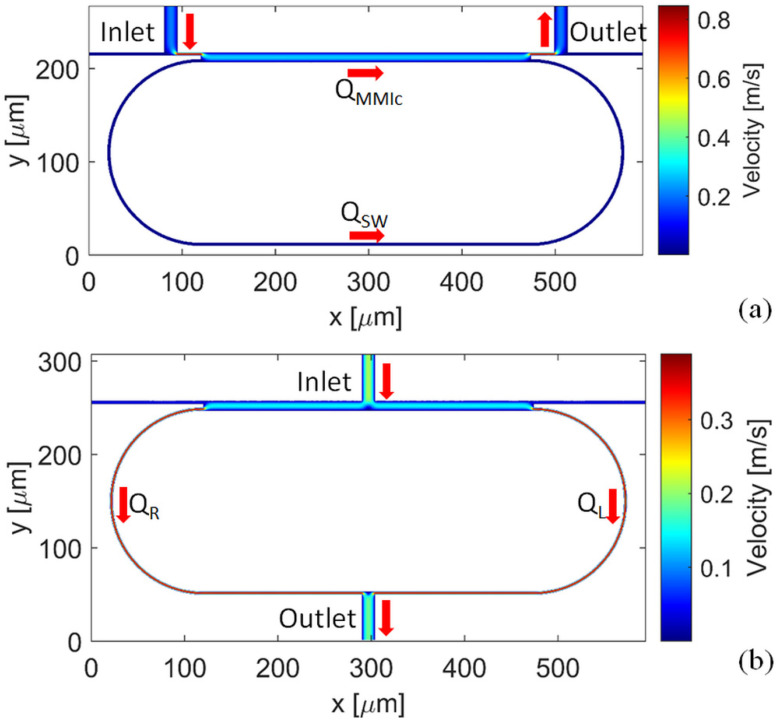
Three-dimensional steady-state flow profile of the (**a**) unbalanced and (**b**) balanced fluidic ring scheme.

**Figure 4 micromachines-13-01028-f004:**
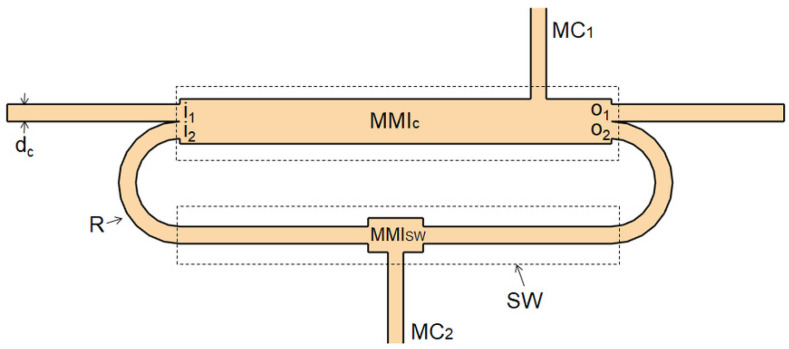
Schematic layout of the proposed optofluidic ring.

**Figure 5 micromachines-13-01028-f005:**
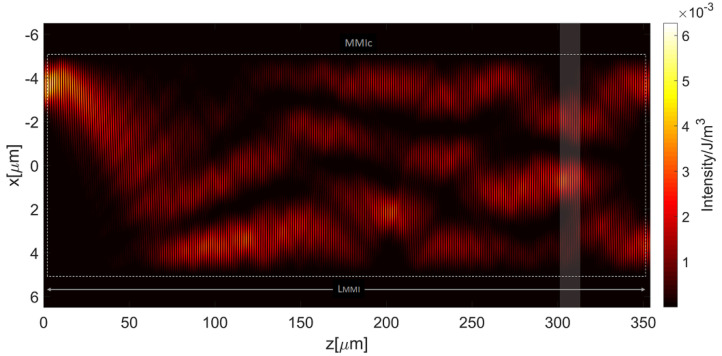
The interference pattern created by multimode interference effect in the ARROW-based multimode interference coupler (MMI_c_). Dotted line indicates the waveguide wall of the MMI_c_.

**Figure 6 micromachines-13-01028-f006:**
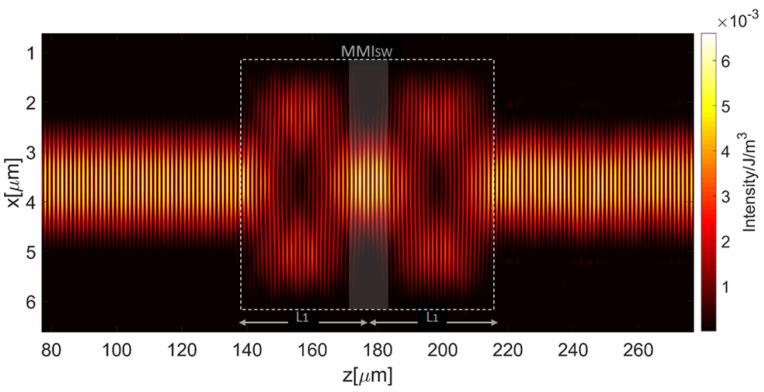
The interference pattern created by multimode interference effect in the MMI_SW_. Dotted line indicates the waveguide wall of the MMI_SW_.

**Figure 7 micromachines-13-01028-f007:**
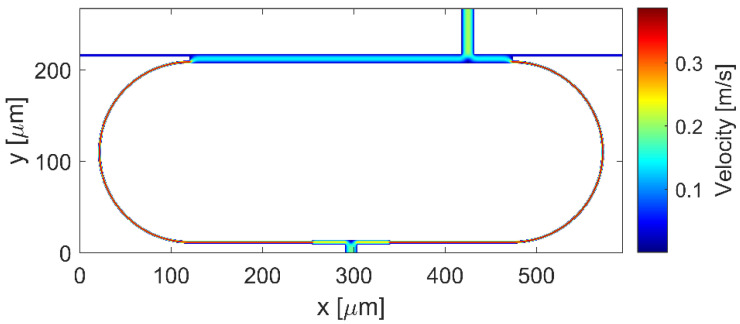
Three-dimensional steady-state flow profile of the optimized ring configuration.

**Figure 8 micromachines-13-01028-f008:**
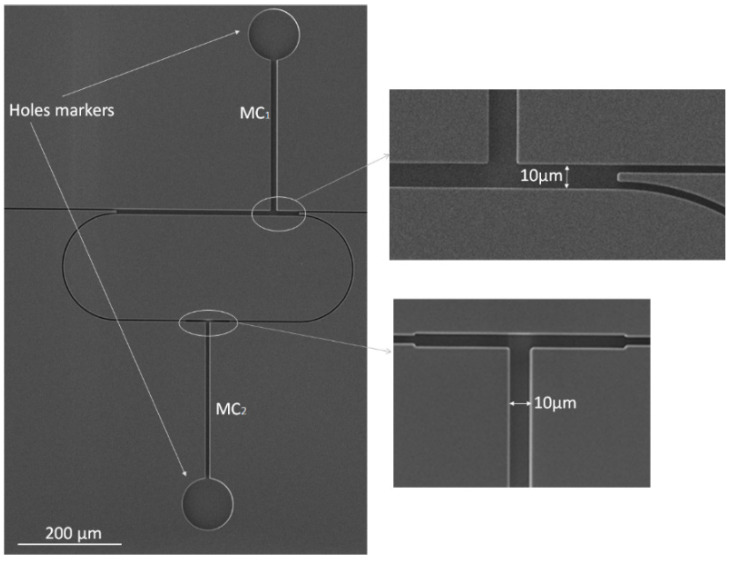
SEM image of the fabricated silicon-based part of the chip.

**Figure 9 micromachines-13-01028-f009:**
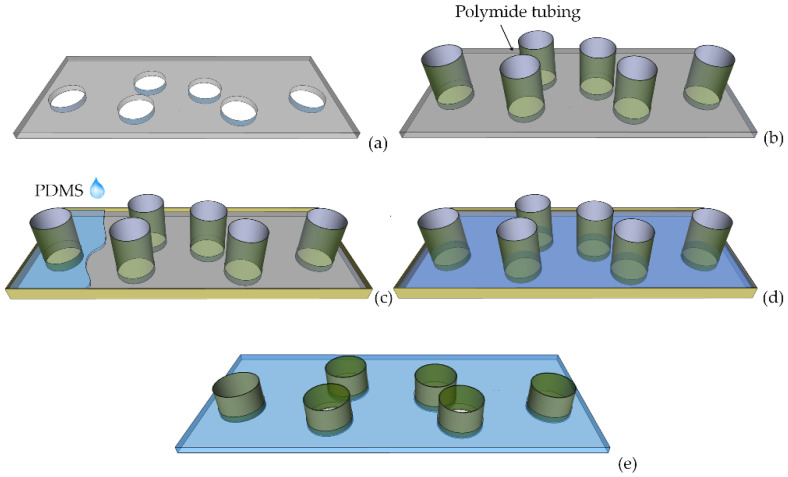
PMMA substrate mold after (**a**) milling holes and (**b**) inserting polymide tubing. (**c**,**d**) PDMS casting on the PMMA mold. (**e**) PDMS layer with integrated polymide tubing.

**Figure 10 micromachines-13-01028-f010:**
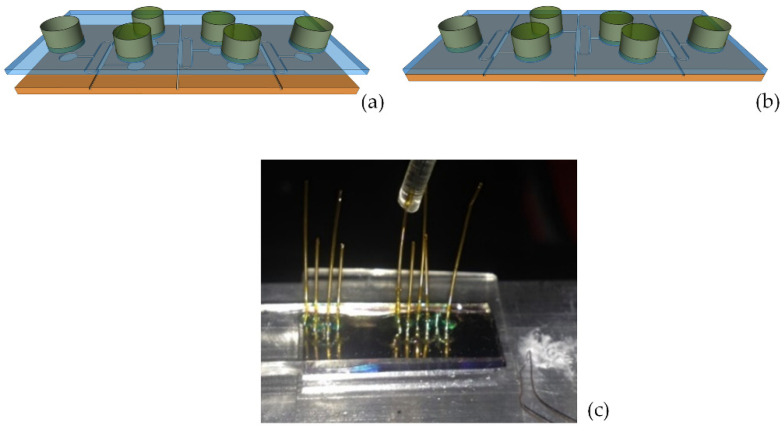
(**a**) Exploded view drawing of the chip. (**b**) Sketch and (**c**) photograph of the final assembled chip.

**Figure 11 micromachines-13-01028-f011:**
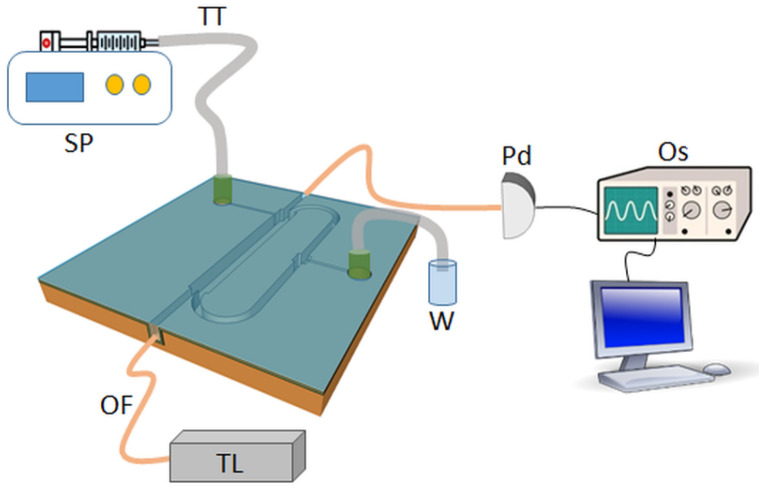
Schematic of the experimental setup. TT, Tygon tubing; SP, syringe pump; OF, optical fiber; Pd, photodiode; TL, tunable laser; W, waste.

**Figure 12 micromachines-13-01028-f012:**
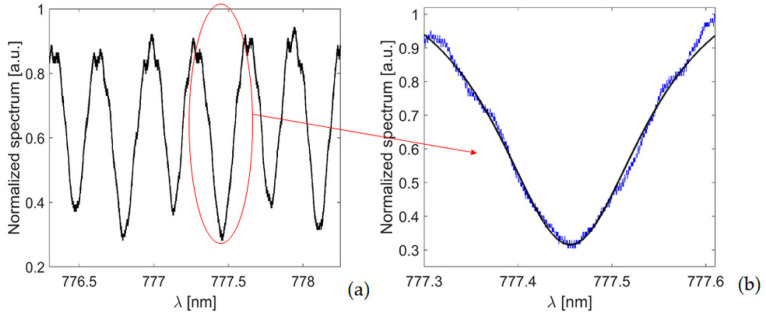
(**a**) Normalized transmission spectrum of the ORR. (**b**) Optical resonance at λ = 777.46 nm and Lorentzian curve fitting (black line).

**Figure 13 micromachines-13-01028-f013:**
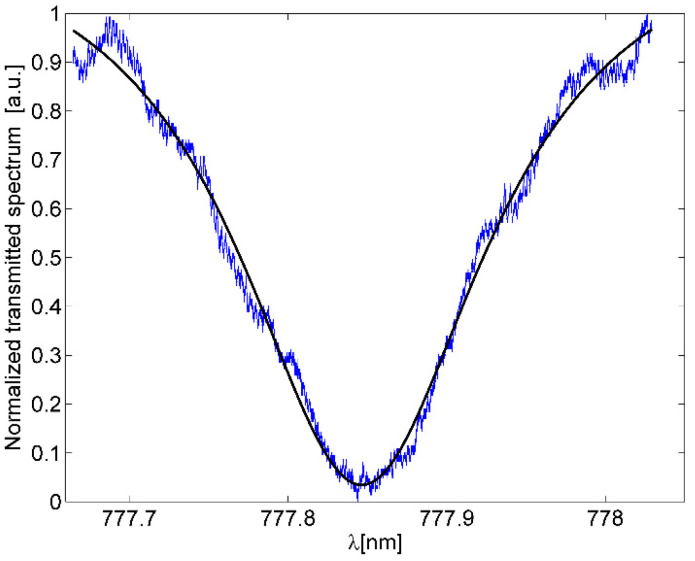
Measured optical resonance and Lorentzian curve fitting (black line) of the proposed ORR_M_.

**Table 1 micromachines-13-01028-t001:** Transmitted power from the MMI_c_ in all the explored cases.

Configuration of the MMI_c_	*P* _1_	*P* _2_
No MC	0.487	0.483
MC at z = 305 µm	0.487	0.483
MC at z = 205 µm	0.435	0.454

**Table 2 micromachines-13-01028-t002:** Transmitted power from the SW in all the explored cases.

Configuration of SW	*P_out,ch_*
No MC_2_	0.970
MC_2_	0.850
MMI_sw_	0.950
MMI_sw_ + MC_2_	0.945

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
