# Peer review of "Planar Optofluidic Integration of Ring Resonator and Microfluidic Channels"

_micromachines, 2022, doi:10.3390/mi13071028_

Round 1

Reviewer 1 Report

Comments:

In this manuscript, the authors designed a planar integrated optofluidic ring resonator with integrated microflulidic channels, considering both the fluidic and optical properties.By carefully selecting the positions and the shape of the input and output, together with the insertion of MMIc and MMIsw parts, a planar integrated optoflulidic ring resonator is optimised, and then frabricated and characterized experimentally. The whole work is innovative and intersting.I think the manuscript can be publised after some minor revisions. 

1) In the experiment, the authors test the Q factor of the ring resontor, can the authors also test the liquid propagation property, for example, how is the volecity and the completeness when the liquid is replaced in the integrated ring resonator. 

2) please correct some grammar errors in English writting, for example, in the sentence "In Figure 4 is shown the proposed layout." , there is no subject in this sentence. Similar errors can be found in the whole manuscript, please correct them.

Reviewer 2 Report

The work by Testa et al. designed and successfully fabricated a planar integrated optofluidic ring resonator with integrated microfluidic channels. Multimode interference pattern was further exploited to minimize the propagation losses caused by waveguides wall disruption. Nevertheless, there are still some points needed to be considered:

1.     In Fig. 8, scale bars should be added in the right panels.

2.     Further experiments are needed to demonstrate the application cases of such designed devices.

3.     Does the flow rate affect the quality factor of the resonator? I expect more discussions here.

Reviewer 3 Report

The authors sought to expand their earlier published work (10.3390/mi7030047). The novelty has worn off and is not merged with an original research article.
In their earlier works they designed 
a liquid core anti-resonant reflecting optical waveguide while, here, they just have integrated it with a microfluidic channel in a LOC system. In my judgement, this work cannot be published on Micromachines journal. 

Round 2

Reviewer 2 Report

All my questions have been resolved. I recommend this paper for publication.

Reviewer 3 Report

The revised paper has not perceptibly improved . As I stated before, due to lack of novelty, this work cannot be published on Micromachines. Poor Integration of a channel with a planar liquid ring resonator is very insufficient.